# Scalable phylogenetic profiling using MinHash uncovers likely eukaryotic sexual reproduction genes

David Moi[1,2,3]*, Laurent Kilchoer[1,2,3], Pablo S. Aguilar [4,5], Christophe Dessimoz[1,2,3,6,7]*

**1** Department of Computational Biology, University of Lausanne, Lausanne, Switzerland, **2** Center for Integrative Genomics, University of Lausanne, Lausanne, Switzerland, **3** SIB Swiss Institute of Bioinformatics, Lausanne, Switzerland, **4** Instituto de Investigaciones Biotecnologicas (IIBIO), Universidad Nacional de San Martín, Buenos Aires, Argentina, **5** Instituto de Fisiología, Biología Molecular y Neurociencias (IFIBYNE-CONICET), Buenos Aires, Argentina, **6** Department of Genetics, Evolution, and Environment, University College London, London, United Kingdom, **7** Department of Computer Science, University College London, London, United Kingdom

* david.moi@unil.ch (DM); christophe.dessimoz@unil.ch (CD)

**Data Availability Statement:** The data is available as supplementary materials. The code is released on GitHub (https://github.com/DessimozLab/HogProf) under an open source license.

## Abstract

Phylogenetic profiling is a computational method to predict genes involved in the same biological process by identifying protein families which tend to be jointly lost or retained across the tree of life. Phylogenetic profiling has customarily been more widely used with prokaryotes than eukaryotes, because the method is thought to require many diverse genomes. There are now many eukaryotic genomes available, but these are considerably larger, and typical phylogenetic profiling methods require at least quadratic time as a function of the number of genes. We introduce a fast, scalable phylogenetic profiling approach entitled HogProf, which leverages hierarchical orthologous groups for the construction of large profiles and locality-sensitive hashing for efficient retrieval of similar profiles. We show that the approach outperforms Enhanced Phylogenetic Tree, a phylogeny-based method, and use the tool to reconstruct networks and query for interactors of the kinetochore complex as well as conserved proteins involved in sexual reproduction: Hap2, Spo11 and Gex1. HogProf enables large-scale phylogenetic profiling across the three domains of life, and will be useful to predict biological pathways among the hundreds of thousands of eukaryotic species that will become available in the coming few years. HogProf is available at https://github.com/DessimozLab/HogProf.

## Author summary

Genes that are involved in the same biological process tend to co-evolve. This property is exploited by the technique of phylogenetic profiling, which identifies co-evolving (and therefore likely functionally related) genes through patterns of correlated gene retention and loss in evolution and across species. However, conventional methods to computing and clustering these correlated genes do not scale with increasing numbers of genomes. HogProf is a novel phylogenetic profiling tool built on probabilistic data structures. It allows the user to construct searchable databases containing the evolutionary history of

**Funding:** This work was funded by a grant by the Novartis Foundation for Medical-Biological Research (#17B111 to CD), by the Swiss National Science Foundation (Grant 183723 to CD), by the Swiss Leading House for the Latin American Region (to CD and PSA), and by the Agencia Nacional de Promoción Científica y Tecnológica (PICT-2017-0854 to PSA). (http://www.stiftungmedbiol.novartis.com, https://cls.unisg.ch/de/forschung/leading-house/funding-instruments/2019-seed-money-grants, http://www.snf.ch, https://www.argentina.gob.ar/ciencia/agencia) The funders had no role in study design, data collection and analysis, decision to publish, or preparation of the manuscript.

**Competing interests:** The authors have declared that no competing interests exist.

hundreds of thousands of protein families. Such fast detection of coevolution takes advantage of the rapidly increasing amount of genomic data publicly available, and can uncover unknown biological networks and guide in-vivo research and experimentation. We have applied our tool to describe the biological networks underpinning sexual reproduction in eukaryotes.

## Introduction

The NCBI Sequence Read Archive (SRA) contains $1.6 \times 10^{16}$ nucleotide bases of data and the quantity of sequenced organisms keeps growing exponentially. To make sense of all of this new genomic information, annotation pipelines need to overcome speed and accuracy barriers. Even in a well-studied model organism such as *Arabidopsis thaliana*, nearly a quarter of all genes are not annotated with an informative gene ontology term [1,2]. One way to infer the function of a gene product is to analyse the biological network it is involved in. Using guilt by association strategies it is possible to infer function based on physical or regulatory interactors. Unfortunately, biological network inference is mostly limited to model organisms and genome scale data is only available through the use of noisy high-throughput experiments.

To ascribe biological functions to these new sequences, most of which originate from non-model organisms, computational methods are essential [reviewed in 3]. Among the computational function prediction techniques that leverage the existing body of experimental data, one important but still underutilised approach in eukaryotes is *phylogenetic profiling* [4]: positively correlated patterns of gene gains and losses across the tree of life are suggestive of genes involved in the same biological pathways.

Phylogenetic profiling has been more commonly performed on prokaryotic genomes than on eukaryotic ones. Perhaps due to the relative paucity of eukaryotic genomes in the 2000s, earlier benchmarking studies observed poorer performance in retrieving known interactions with eukaryotes than with Prokaryotes [5–7]. The situation today is considerably different; the GOLD database [8] tracks over 6000 eukaryotic genomes. Multiple successful applications of phylogenetic profiling in eukaryotes have been published in recent years. For example, they have been used to infer small RNA pathway genes [9], the kinetochore network [10], ciliary genes [11], or homologous recombination repair genes [12].

Large-scale phylogenetic profiling with complex eukaryotic genomes is computationally challenging since most state-of-the-art phylogenetic profiling methods typically scale at least quadratically with the number of gene families and linearly with the number of genomes. As a result, most mainstream phylogenomic databases, such as Ensembl [13], EggNOG [14], OrthoDB [15], or OMA [16] do not provide phylogenetic profiles. One available resource is STRING [17], a protein interaction focused database which integrates multiple channels of evidence to support each interaction. The links between profiles STRING offers are obtained using SVD-phy [18] which represents profiles as bit-score distances between all proteins present in a given proteome and their closest homologues in all of the genomes included in the analysis. Dimensionality reduction is applied to the matrix to remove signals coming from the species tree and the profiles are clustered to infer interactions. In STRING, this is implemented with their set of 2031 organisms for which profile distance matrices are precalculated and incorporated into their network inference pipeline. Although this approach captures information on the distribution of extant distances, it does not reconstitute the evolutionary history of protein families and may lack information relative to duplication and loss events. Furthermore, as we show in the *Methods section*, the truncated Singular Value Decomposition approach does not scale well beyond a few genomes at a time.

To construct profiles representing groups of homologues, some pipelines resort to all-vs-all sequence similarity searches to derive orthologous groups and only count binary presence or absence of a member of each group in a limited number of genomes [19,20] or forgo this step altogether and ignore the evolutionary history of each protein family, relying instead on co-occurrence in extant genomes [21]. Other tree-based methods infer the underlying evolutionary history from the presence of extant homologues [22].

Here, we introduce a scalable approach which combines the efficient generation of phylogeny-aware profiles from hierarchical orthologous groups with ultrafast retrieval of similar profiles using locality sensitive hashing.

Most phylogenetic profiling methods consist of two steps: creating a profile for each homologous or orthologous group, and comparing profiles. When they were first implemented, profiles were constructed as binary vectors of presence and absence across species [4]. Since then, variants have been proposed, which take continuous values [10]—such as alignment scores with the gene of a reference species [12]—or which count the number of paralogs present in each species. Yet other variants convey the number of events on branches of the species tree [7].

In our pipeline, we leveraged the already existing OMA orthology inference algorithm to provide the input data to create our profiles. The OMA database describes the orthology relationships among all protein coding genes of currently 2288 cellular organisms (1674 bacteria, 152 archaea, and 462 eukaryotes). Within eukaryotes, OMA includes 188 animals, 135 fungi, 57 plants, and 82 protists and has been benchmarked and integrated with other proteomic and genomic resources [16]. One core object of this database is the Hierarchical Orthologous Group (HOG) [23]. Each HOG contains all of the descendants of a single ancestor gene. When a gene is duplicated during its evolution, the paralogous genes and the descendants of the orthologue are contained in separate subhogs which describe their lineage back to their single ancestor gene (hence the hierarchical descriptor).

We captured the evolutionary history of each HOG in enhanced phylogenies and encoded them in probabilistic data structures (Fig 1). These are used to compile searchable databases to allow for the retrieval of coevolving HOGs with similar evolutionary histories and compare the similarity of two HOGs. The two major components of the pipeline that are responsible for constructing the enhanced phylogenies and calculating probabilistic data structures to represent them are pyHam [24] and Datasketch [25], respectively. The combination of these two tools now allows for the main innovation of our pipeline: the efficient exploration and clustering of profiles to study known and novel biological networks.

Currently, existing profiling pipelines are limited with respect to the computational power required to cluster profiles using their respective distance metrics. Due to this bottleneck, profiling efforts are typically focused on reconstructing pathways with known interactors using existing annotations and evidence rather than being used as an exploratory tool to search for new interactors and reconstituting completely unknown networks.

The tool we have developed leverages the properties of MinHash signatures to allow for the selection of clade subsets and for clade weightings in the construction of profiles and make it possible to build profiles with the complete set of genomes contained in OMA. We show that the method outperforms other phylogeny-based methods [19,26,27], and illustrate its usefulness by retrieving biologically relevant results for several genes of interest. Because the method is unaffected by the number of genomes included and scales logarithmically with the number of hierarchical orthologous groups added, it will efficiently perform with the exponentially growing number of genomes as they become available.

The code used to generate the results in this manuscript are available at https://github.com/DessimozLab/HogProf.

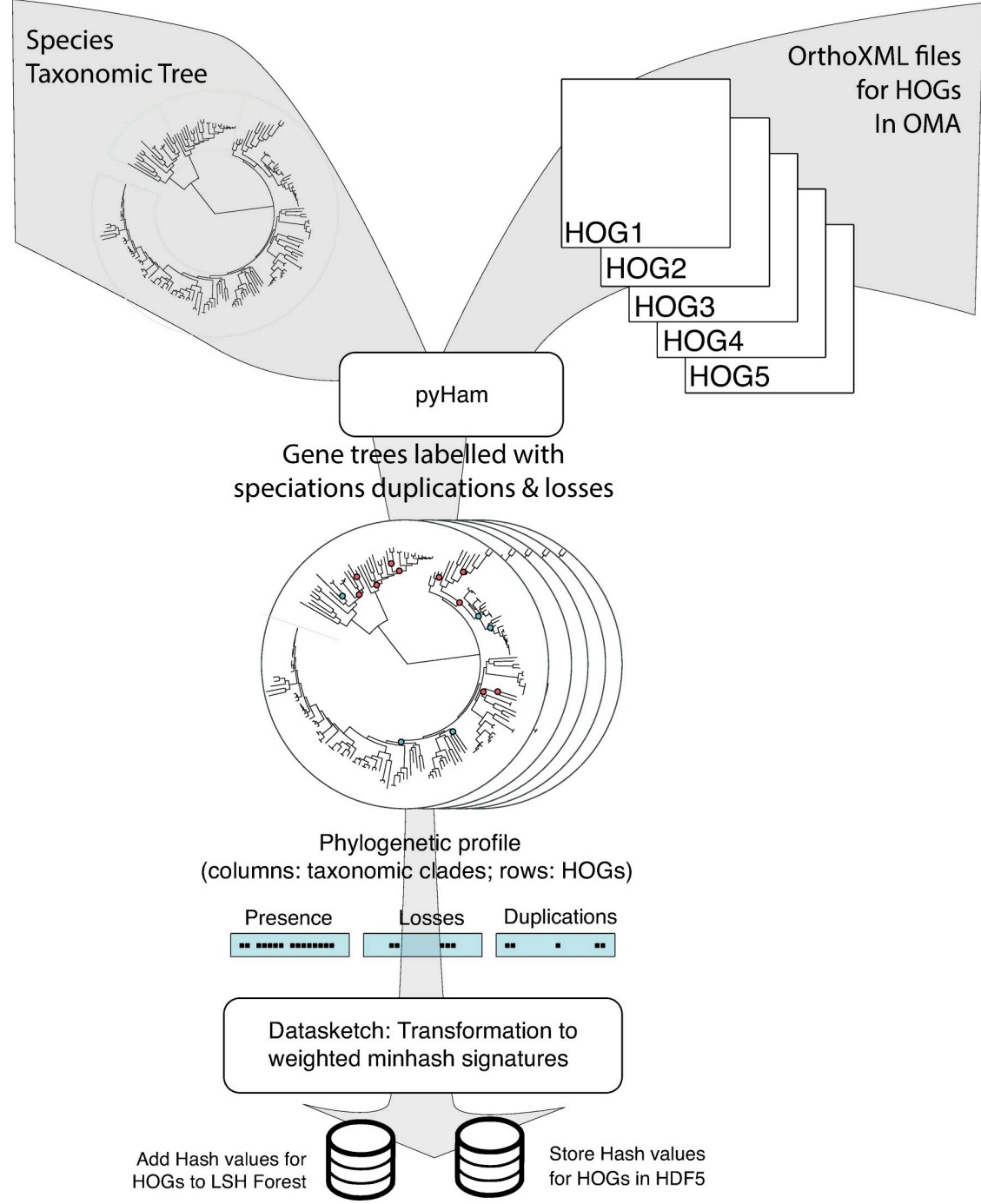

**Fig 1. Diagram summarizing the different steps of the pipeline to generate the LSH Forest and hash signatures for each HOG.**
The labelled phylogenetic trees generated by pyHam are converted into phylogenetic profiles and used to generate a weighted
MinHash signature with Datasketch. The hash signatures are inserted into the LSH Forest and stored in an HDF5 file.

## Results

In the following sections we first compare our profiling distance metric against other profile
distances in order to characterize the Jaccard hash estimation's precision and recall character-
istics. Following this quantification, we show our pipeline's capacity in reconstituting a well

known interaction network as well as augmenting it with more putative interactors using its search functionality. Finally, to illustrate a typical use case of our tool, we explore a poorly characterized network.

## Accuracy of predicted phylogenetic profiles in an empirical benchmark

We compared the performance of our profiling metric to existing profile distances using benchmarking data available in Ta *et al.* [19]. In that benchmark, the true positive protein-protein interactions (PPIs) were constructed using data available from CORUM [28] and the MIPS [29] databases for the human and yeast interaction datasets. True negatives were constructed by mixing proteins known to be involved in different complexes. The dataset is balanced with 50% positive and 50% negative samples. Using their Uniprot identifiers, these interaction pairs were mapped to their respective HOGs and their profiles were compared using the hash-based Jaccard score estimate. The comparison below shows HogProf alongside other profiling distance metrics that are considerably more computationally intensive, including the Enhanced Phylogenetic Tree (EPT) metric shown in Ta *et al.* [19]. Yet, our approach outperformed these previous methods, yielding the highest Area Under the Curve for both yeast and human datasets (Fig 2, Table 1).

## Recovery of a canonical network: the kinetochore network

To further validate our profiling approach on a known biological network, we used our pipeline to replicate previous work shown in van Hooff et al. [10]. Their analysis focuses on the

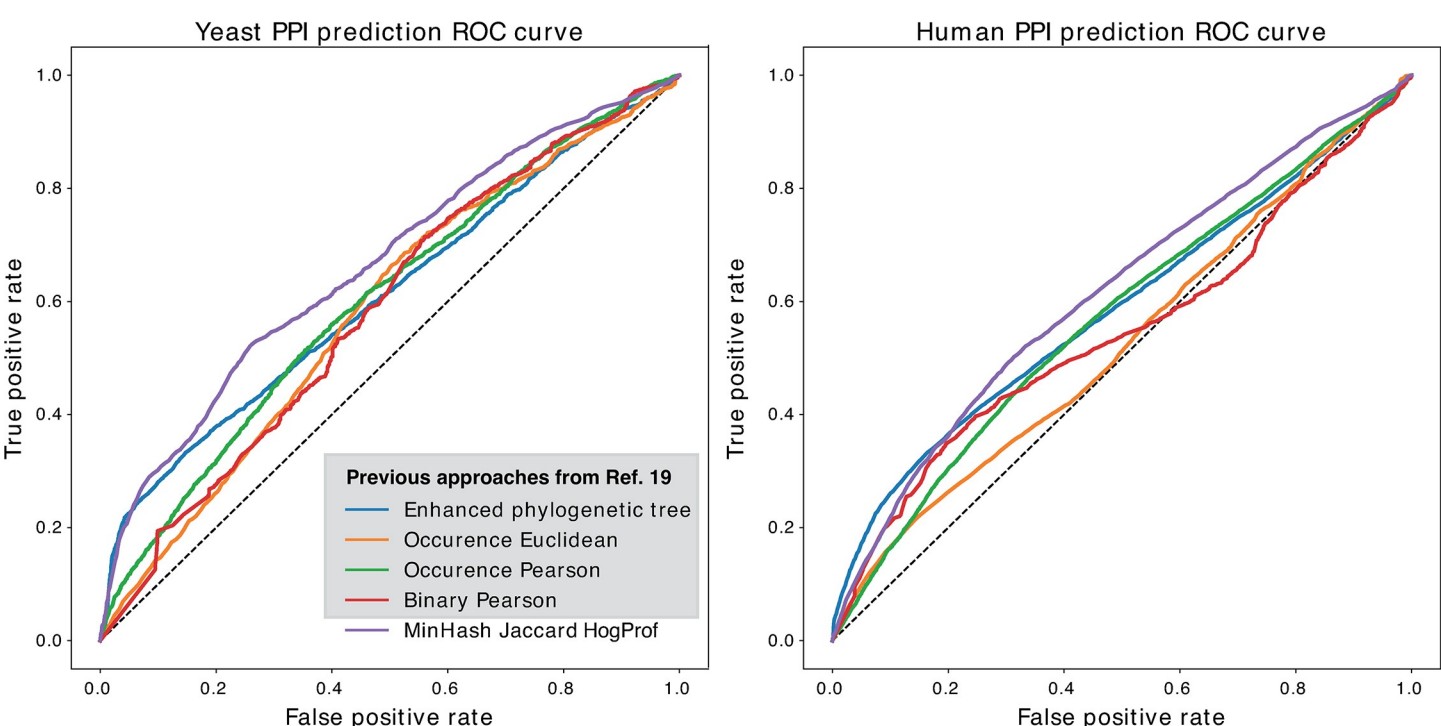

**Fig 2. ROC curves for all profiling methods. a.** Yeast protein-protein interactions. Our method (MinHash Jaccard HogProf), performs best overall, but when high precision is required, Enhanced phylogenetic Tree [19] is still slightly more accurate. **b.** Human protein-protein interactions. Jaccard Hash HogProf performs better than all metrics overall but again, when high precision is required, EPT score is still slightly more accurate. Binary Pearson refers to a distance using binary vectors and Pearson correlation described in [26]. Occurence Euclidean and Occurence Pearson refer to the occurence profiles with Euclidean distance and Pearson correlation as described in [27].

**Table 1. AUC values for Profiling distance metrics.**

| Metric | AUC Yeast | AUC Human |
|---|---|---|
| Jaccard Hash | 0.6634 | 0.6155 |
| EPT | 0.6104 | 0.5875 |
| BIN PS | 0.5840 | 0.5463 |
| OCC ED | 0.5829 | 0.5268 |
| OCC PS | 0.6028 | 0.5714 |

evolutionary dynamics of the kinetochore complex, a microtubule organizing structure that was present in the last eukaryotic common ancestor (LECA) and has undergone many modifications throughout evolution in each eukaryotic clade where it is found. Its modular organization has allowed for clade-specific additions or deletions of modules to the core complex which remains relatively stable. This modular organisation and clade-specific emergence of certain parts of the complex make it an ideal target for phylogenetic profiling analysis.

We show that our MinHash signature comparisons are also capable of recovering the kinetochore complex organisation. After considering just the HOGs for the families used in van Hooff et al. [10], we augmented their set of profiles using LSH Forest [30] to retrieve interactors that may also be involved in the kinetochore (and the also included anaphase promoting complex (APC)) networks which have not been catalogued by these authors. Using the Gene Ontology (GO) terms [31] of all proteins returned in our searches for novel interactors, we were able to identify proteins with specific functions we would expect to be related to our network of interest.

In their work, van Hooff et al. [10] used pairwise Pearson correlation coefficients between the presence and absence vectors of the various kinetochore components to recompose the organisation of the complex. Their profiles were constructed using the proteomes of a manually selected set of 90 organisms with manually curated profiles corresponding to each component of the complex. After establishing a distance kernel, they clustered the profiles and were able to recover known sub-components of the complex using just evolutionary information. Using our hash-based Jaccard distance metric in an all-vs-all comparison between the HOGs corresponding to each of these protein families, we were also able to recover the main modules of the kinetochore complex with a similar organisation to the one defined by van Hooff et al. The color clustering in Fig 3 corresponds to their original manual definition of these different subcomplex modules. We observe that the distance matrices generated by each profiling approach are correlated (with Spearman correlation of 0.268 (p < 1e-100) and Pearson correlation of 0.364 (p < 1e-100)) and are recovering similar evolutionary signals despite their construction using different methods.

The All-vs-All comparison of the profiles revealed several well defined clusters in both studies including the Dam-Dad-Spc19 and CenP subcomplexes. Unlike the Van Hootf et al. approach, HogProf profiles were constructed alongside all other HOGs in OMA and were not curated before being compared. With only the initial information of which proteins were in the complex, we mapped them to their corresponding OMA HOGs and, with this example, demonstrated the ability to reconstruct any network of interest or construct putative networks using the search functionality of our pipeline with minimal computing time. It should be noted that the quality of the OMA HOGs used to construct the enhanced phylogenies and hash signatures directly influences our ability to recover complex organisation.

To illustrate the utility of the search functionality of our tool, we used the profiles known to be associated with the kinetochore complex to search for other interactors. All HOGs corresponding to the protein families used to analyse the kinetochore evolutionary dynamics in van

 Scalable phylogenetic profiling uncovers likely eukaryotic sexual reproduction genes

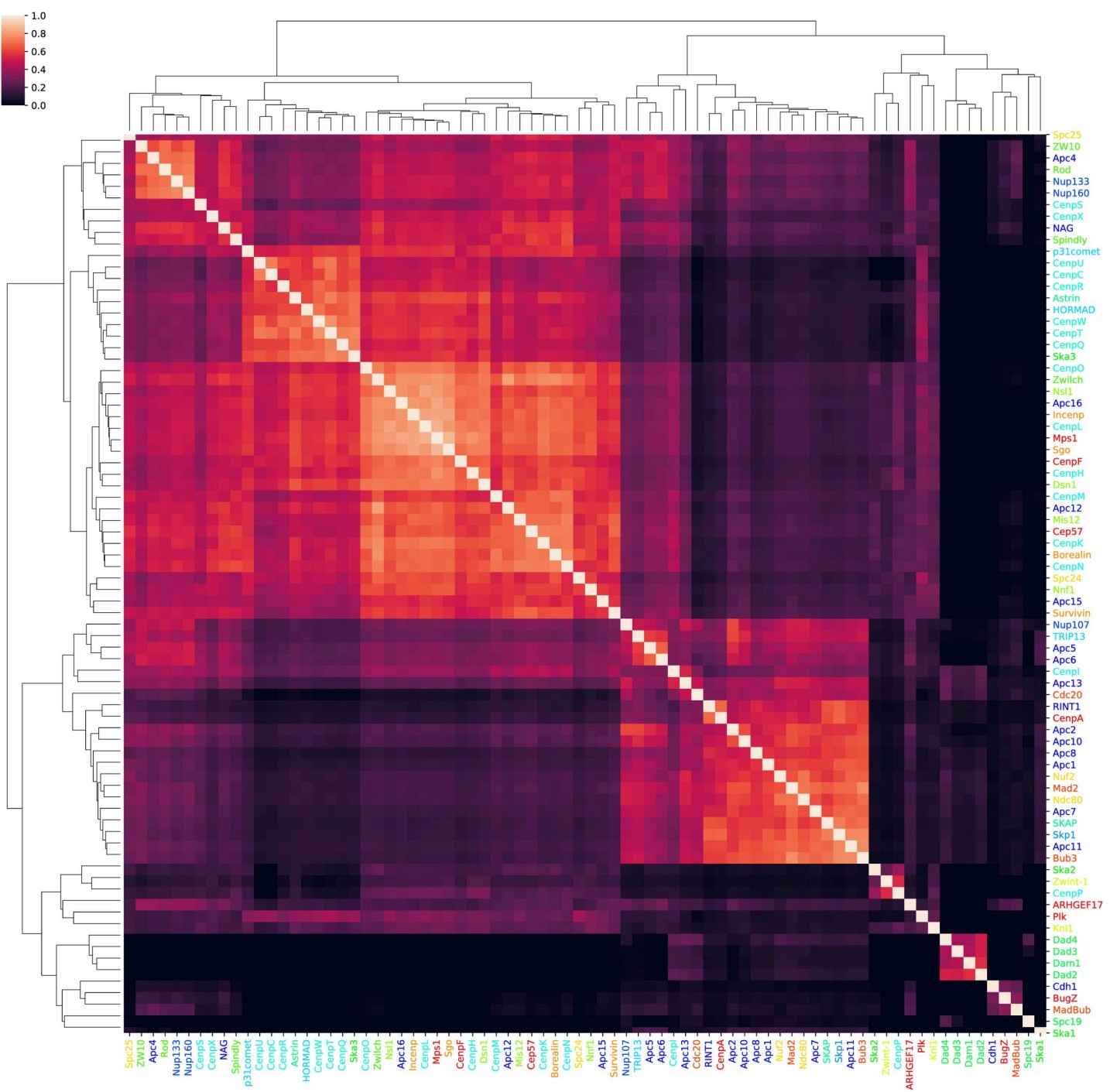

**Fig 3. Recovery of kinetochore and APC complexes.** After mapping each of the protein families presented in Van Hooff et al. [10] to their corresponding HOG, a distance matrix was constructed by comparing the Jaccard hash distance between profiles using HogProf. Name colors in the rows and columns of the matrix correspond to the kinetochore and APC subcomplex components as defined manually using literature sources [10].

Hooff et al. [10] were used as queries against an LSH Forest containing all HOGs in OMA. By performing an all-vs-all comparison of the minhash signatures of the queries and returned results, a Jaccard distance matrix was generated showing potential functional modules associated with each known component of the kinetochore and APC complexes.

To verify that the results returned by our search were not spurious, we performed GO enrichment analysis of the returned HOGs that were not part of the original set of queries but appeared to be coevolving closely with known kinetochore components. Given the incomplete nature of GO annotations ["open world assumption", 32], many of these proteins may actually be involved in the kinetochore interaction network but this biological function could be still undiscovered. However, even with this limitation, salient annotations relevant to the kinetochore network were returned in the search results (Table 2 and S1 Data). The identifiers of all protein sequences contained in the HOGs returned by the search results were compiled and the GO enrichment of each cluster shown in Fig 4 was calculated using the OMA annotation corpus as a background. The enrichment results were manually parsed and salient annotations related to HOGs were selected to be reviewed further in the associated literature to check for the association of the search result with the query HOG (Table 2).

For instance, our search identified TACC3, which is known to be part of a structural stabilizer of kinetochore microtubules tension although it does not directly interact with the kinetochore complex [36]. ESCO2, a cohesin N-acetyltransferase needed for proper chromosome segregation during meiosis also plays a role in kinetochore-microtubule attachments regulation during meiosis [35]. While these results are certainly promising, many of the unannotated proteins returned by our search likely contain more regulatory, metabolic and physical interactors which may prove to be interesting experimental targets.

## Search for a novel network

Typical research use cases for profiling often involve uncharacterized protein families acting within poorly studied neworks. In this section we present search results for three HOGs known to be involved in the processes of meiosis, syngamy and karyogamy. Despite the ubiquitous nature of sex and its probable presence in LECA [37], the protein networks involved in each part of these processes have limited experimental data available, even in model organisms. Some key protein families involved in each step are known to have evolutionary patterns indicating an ancestral sequence in the LECA with subsequent modifications and losses [37]. The three following sections detail the returned results of the phylogenetic profiling pipeline with the Hap2, Gex1 and Spo11 families which all share this evolutionary pattern and are known to be critical for the process of gamete fusion, nuclear fusion and meiotic recombination, respectively. The proteins contained in the top 100 HOGs returned by the LSH Forest were analyzed for GO enrichment using all OMA annotations as a background. Due to the presence of biases in the GO annotation corpus [38] we have also chosen to show the number of proteins annotated with each biological process selected from the enrichment out of the total number of annotated proteins.

## Query with Hap2

The Hap2 protein family has been shown to catalyze gamete membrane fusion in many eukaryotic clades and shares structural homology with viral and somatic membrane fusion

**Table 2. Manually curated biologically relevant search results for interactors coevolving with van Hooff *et al.*'s kinetochore and APC selected protein families [10].** Protein families returned within clusters containing query HOGs are listed with their pertinent annotation and literature. This is a non-exhaustive summary of some selected results. The full enrichment results are available as S1 Data.

| Cluster | Result | GO Term | Citation |
|---------|--------|---------|----------|
| APC1 | CFAP157 | GO:0035082 axoneme assembly | [33] |
| APC12 | C2CD3 | GO:0061511 centriole elongation | [34] |
| CenpQ | ESCO2 | GO:0007059 chromosome segregation | [35] |
| KNL1 | TACC3 | GO:0007091 metaphase/anaphase transition of mitotic cell cycle | [36] |

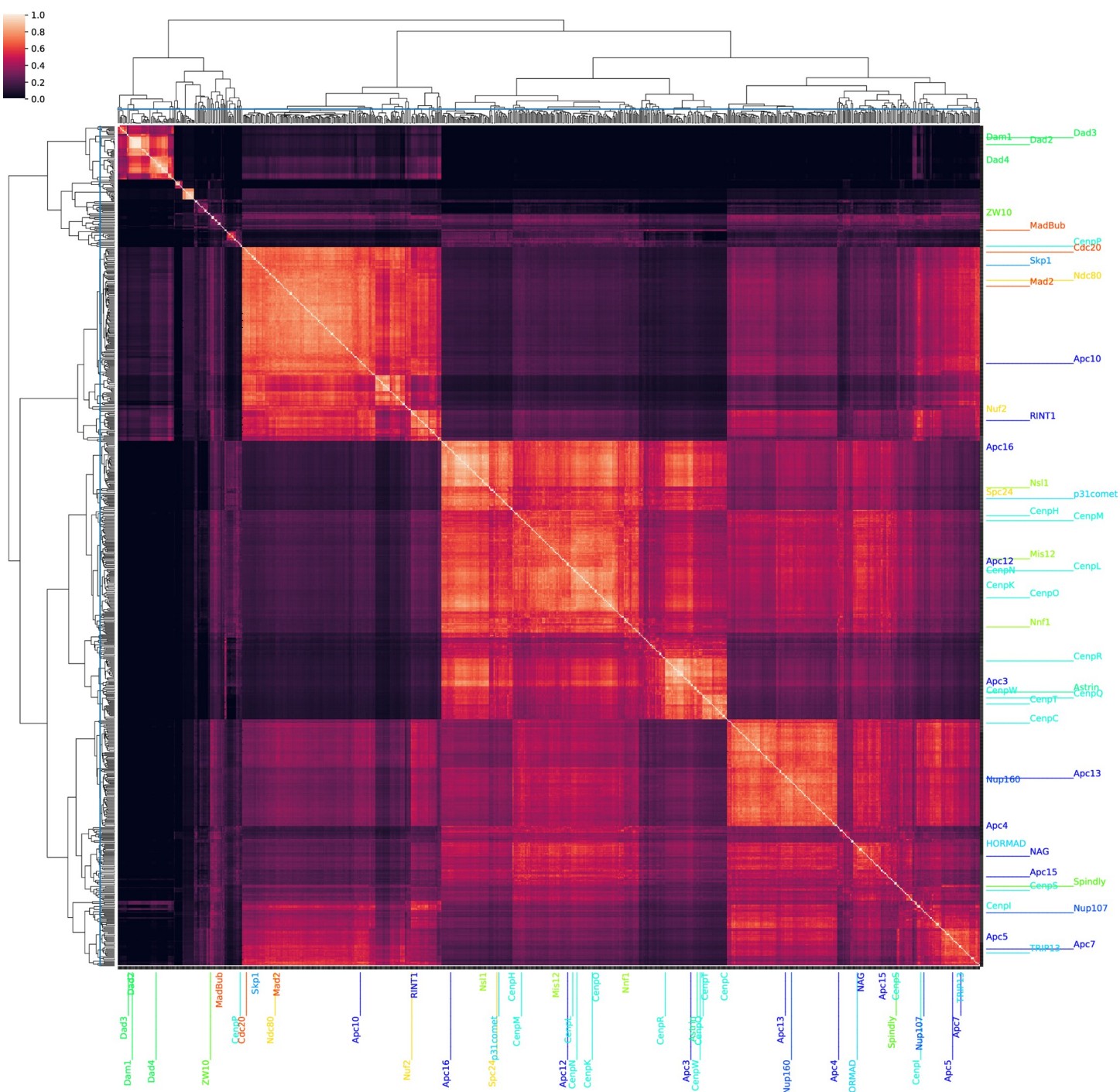

**Fig 4. Putative novel components of the kinetochore and APC complexes.** The profiles associated with all HOGs mapping to known kinetochore components shown in Fig 3 were used to search the LSH Forest and retrieve the top 10 closest coevolving HOGs resulting in a list of 871 HOGs including the queries from the original complexes. The Jaccard distance matrix is shown between the hash signatures of all query and result HOGs. UPGMA clustering was applied to the distance matrix rows and columns. Labelled rows and columns correspond to profiles from the starting kinetochore dataset [10]. A cutoff hierarchical clustering distance of 1.3 was manually chosen (blue lines) to limit the maximum cluster size to less than 50 HOGs. This cutoff resulted in a total of 142 clusters of HOGs used for GO enrichment to identify functional modules. The coloring of the protein family names to the right and below the matrix is identical to the complex related coloring shown in Fig 3.

**Table 3. Manually curated biologically relevant enriched GO terms from returned results.** The query sequence Hap2 is UniProt entry F4JP36 with OMA identifier ARATH26614 belonging to OMA HOG:0406399. The full enrichment results are available in the S2 Data.

| Term | Biological process | P-value | N-proteins |
|------|-------------------|---------|-----------|
| GO:0006338 | chromatin remodeling | 9.72e-54 | 61/3426 |
| GO:0048653 | anther development | 1.69e-35 | 17/3426 |
| GO:0009793 | embryo development ending in seed dormancy | 2.88e-13 | 15/3426 |
| GO:0051301 | cell division | 6.88e-16 | 5/3426 |

proteins [39–42]. A subset of the GO enrichment of the search results for the top 100 coevolving HOGs are shown below in Table 3.

One particular family of interest which was returned in our search results is already characterized in angiosperms: LFR or leaf and flower related [43]. This protein family is required for the development of reproductive structures in flowers and serves as a master regulator of the expression of many reproduction related genes, but its role in lower eukaryotes remains undescribed despite its broad evolutionary conservation.

## Query with Gex1

The nuclear fusion protein Gex1 is present in many of the same clades as Hap2, with a similar spotty pattern of absence across eukaryotes and a phylogeny indicating a vertical descent from LECA [44]. A subset of the GO enrichment of the search results for the top 100 coevolving HOGs are shown below in Table 4.

Gex1 has been shown to be involved in gamete development and embryogenesis [45] and therefore GO terms 0022619 and 0009553 are applied to this protein. Thus proteins that Hog-Prof identified as co-evolving with Gex1 and sharing these GO terms can be considered potential Gex1 interactors.

One search result of particular interest is a protein family which goes by the lyrical name of parting dancers (PTD). PTD belongs to a family that has been characterized in *Arabidopsis thaliana* and budding and fission yeast, and is known to be required in reciprocal homologous recombination during meiosis [46]. Our search shows that Gex1 co-evolved closely with PTD, a protein known to be involved in preparing genetic material for its eventual merger with another cell's nucleus.

## Query with Spo11

The Spo11 helicase is involved in meiosis by catalyzing DNA double stranded breaks (DSBs) triggering homologous recombination. Spo11 is highly conserved throughout eukaryotes and

**Table 4. Manually curated biologically relevant enriched GO terms from returned results.** The query sequence Gex1 is UniProt identifier Q681K7 with OMA identifier ARATH38809 belonging to OMA HOG:0416115. The full enrichment results are available as S3 Data.

| GO Term | P-value | N-Proteins |
|---------|---------|-----------|
| GO:0042753 positive regulation of circadian rhythm | 2.12e-285 | 113/2685 |
| GO:0048364 root development | 7.81e-125 | 70/2685 |
| GO:0051726 regulation of cell cycle | 1.22e-92 | 99/2685 |
| GO:0000712 resolution of meiotic recombination intermediates | 1.65e-47 | 26/2685 |
| GO:0007140 male meiotic nuclear division | 1.19e-39 | 26/2685 |
| GO:0009553 embryo sac development | 1.43e-28 | 17/2685 |
| GO:0022619 generative cell differentiation | 3.59e-18 | 5/2685 |

**Table 5. Manually curated biologically relevant enriched GO terms from returned results.** The query sequence Spo11-1 is UniProt identifier Q9M4A2 with OMA identifier ARATH19148 belonging to OMA HOG:0605395.

| GO Term | P-value | N-Proteins |
|---|---|---|
| GO:0000737 DNA catabolic process, endonucleolytic | 0.00e+00 | 415/20562 |
| GO:0043137 DNA replication, removal of RNA primer | 0.00e+00 | 353/20562 |
| GO:0006275 regulation of DNA replication | 0.00e+00 | 552/20562 |
| GO:0006302 double-strand break repair | 8.11e-242 | 285/20562 |
| GO:0007292 female gamete generation | 2.71e-184 | 136/20562 |
| GO:0022414 reproductive process | 1.66e-93 | 127/20562 |

homologues are present in almost all clades [47]. A subset of the GO enrichment of the search results for the top 100 coevolving HOGs are shown below in Table 5.

It is encouraging to find that Spo11, the trigger of meiotic DSBs, has co-evolved with other families involved in the inverse process of repairing the DSBs and finishing the process of recombination (Table 5). Other identified HOGs contain annotations such as gamete generation and reproduction also focusing on processes that result in cellular commitment to a gamete cell fate through meiosis. Proliferating cell nuclear antigen or PCNA [48] was also retrieved by our search. This ubiquitous protein family is an auxiliary scaffold protein to the DNA polymerase and recruits other interactors to the polymerase complex to repair damaged DNA, making it an interesting candidate for a potential physical interactor with Spo11.

## A broader search for the reproductive network

A more in-depth treatment of the evolutionary conservation of gamete cell fate commitment and mating is available in previous publications [37,44,49–53]. Using these sources, a list of broadly conserved protein families known to be involved in sexual reproduction were compiled (S4 Data) to be used as HOG queries to the LSH Forest to retrieve the top 10 closest coevolving HOGs. The hash signatures of the queries and results were compiled and used in an all-vs-all comparison to generate a Jaccard distance matrix.

The all-vs-all comparison of the Jaccard distances between these returned HOGs reveals clusters of putative interactors co-evolving closely with specific parts of the sexual reproduction network (Fig 5). Manual analysis of GO enrichment results revealed several sexual reproduction-related proteins which are summarized in Table 6. In addition to annotated protein sequences and HOGs, many unannotated, coevolving HOGs were found.

Particularly for biological processes as complex and evolutionarily diverse as sexual reproduction, GO annotations are, unsurprisingly, incomplete. Fortunately, our profiling approach is successful in identifying protein families with similar evolutionary patterns that have already been characterised and are directly relevant to sexual reproduction (Table 6). By considering the uncharacterized or poorly characterized families at the sequence and structure level, we may be able to predict their functions and reconstitute their local interactome. Our ultimate goal is to guide *in vivo* experiments to test and characterize these targets within the broader context of eukaryotic sexual reproduction.

This example related to the ancestral sexual reproduction network illustrates the utility of the LSH Forest search functionality and OMA resources in exploratory characterization of poorly described networks. The interactions presented above (Table 6) only represent our limited effort to manually review literature to highlight potentially credible interactions detected by our pipeline. Again, as was the case with our kinetochore and APC related searches, several interactions might not appear obvious on their face. For example, SPC72 and MID2 are both involved in meiotic processes but localized to different parts of the cell (centriole and plasma

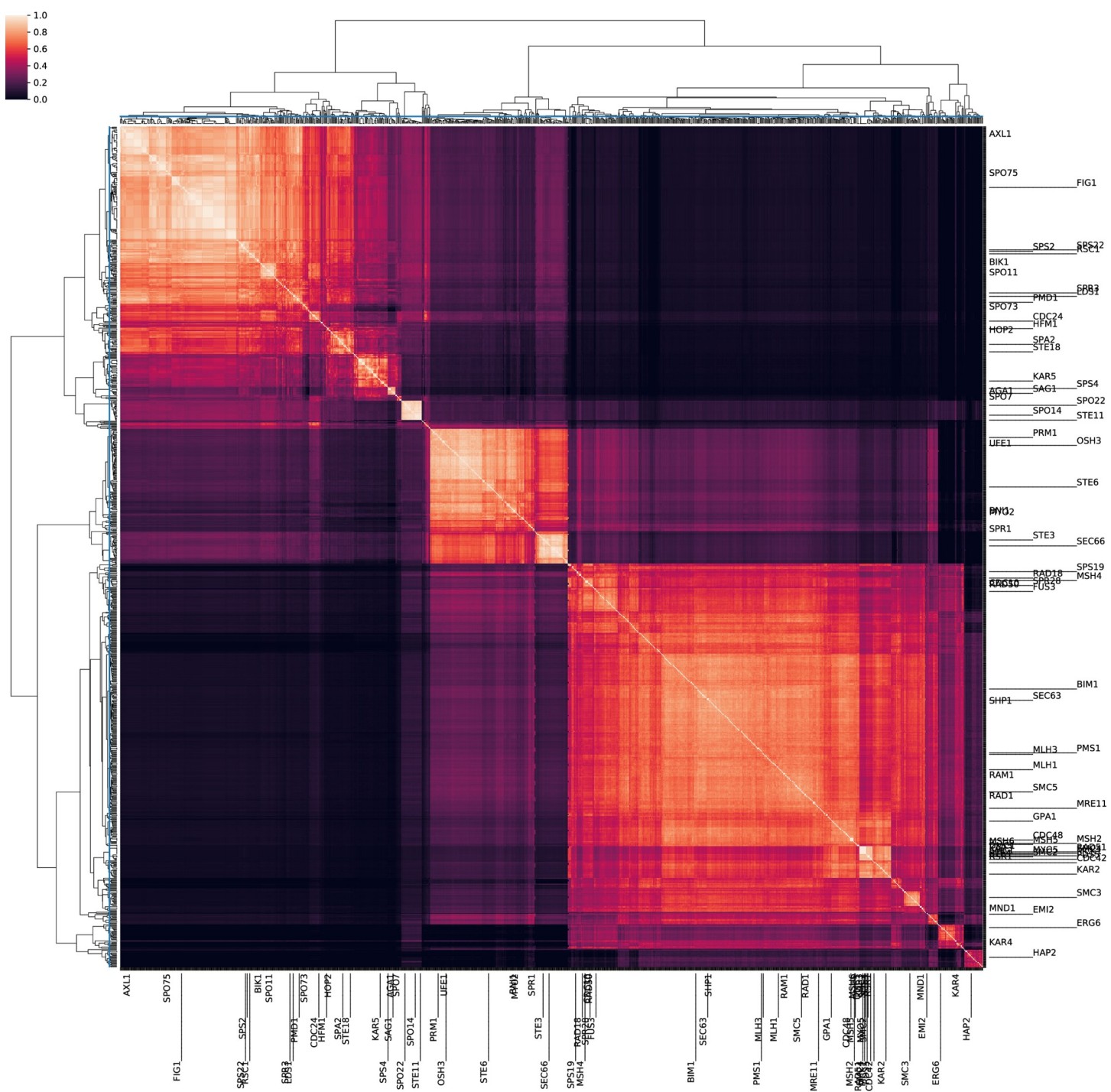

**Fig 5. HogProf's reproductive network.** A list of proteins known to be involved in sexual reproduction was compiled and mapped to OMA HOGs. These queries were used to search for the 20 closest coevolving HOGs in an LSH forest containing all HOGs in OMA. A Jaccard kernel was generated by performing an All vs All comparison of the Hash signatures of search results and queries. UPGMA clustering was performed on the rows and columns of the kernel. A cutoff distance of .995 (blue lines) was manually chosen to limit cluster sizes to less than 50 HOGs. This generated a total of 215 clusters of HOGs. Names for queries are shown with *Saccharomyces cerevisiae* gene names (apart from Hap2 which is not present in fungi).

**Table 6. Manually curated biologically relevant putative interactors from sexual reproduction search results.** Protein families within clusters containing query HOGs are listed with their pertinent annotation and literature. GO enrichment results of clusters containing one or more queries were analyzed manually. Full enrichment results are available as S5 Data.

| Cluster | Result | GO Term | Citation |
|---|---|---|---|
| REC8 | NSE4 | GO:0030915 Smc5-Smc6 complex | [54] |
| SPC72 | MID2 | GO:0000767 cell morphogenesis involved in conjugation | [55] |
| SPO71 | LES2 | GO:0031011 Ino80 complex | [56,57] |
| SHC1, SPO16 | POG1 | GO:0000321 re-entry into mitotic cell cycle after pheromone arrest | [58,59] |

membrane, respectively). However, it has been shown that microtubule organization and membrane integrity sensing pathways do show interaction during gamete maturation [60].

## Discussion

We introduced a scalable system for phylogenetic profiling from hierarchical orthologous groups. The ROC and AUC values shown using an empirical benchmark (*Results* section) indicate that the MinHash Jaccard score estimate between profiles has slightly better performance than previous tree and vector based metrics, while also being much faster to compute. This is remarkable in that one typically expects a trade-off between speed and accuracy, which does not appear to be the case here. We hypothesise that the error introduced by the MinHash approximation is compensated by the inclusion of an unprecedented amount of genomes and taxonomic nodes in the labelled phylogenies used to construct the profiles.

Furthermore, while our MinHash-derived Jaccard estimates are able to capture some of the differences between interacting and non-interacting HOGs, their unique strength lies in the fast recovery of the top k closest profiles within an LSH Forest. Once these profiles are recovered, the inference of submodules or network structure can be refined using other, potentially more compute intensive methods, on this much smaller subset of data.

We have shown that HogProf is able to reconstitute the modular organisation of the kinetochore, as well as increase the list of protein families interacting within the network with several known interactors of the kinetochore and the APC. As for the other HOGs returned in these searches, our results suggest that some are yet unknown interactors involved in aspects of the cell cycle or ciliary dynamics. Likewise, our attempt at retrieving candidate members of the sexual reproduction network recapitulated many known interactions, while also suggesting new ones.

The current paradigm for exploring interaction or participation in different biological pathways across protein families relies heavily on data integration strategies that take into account heterogenous high-throughput experiments and knowledge found in the literature. Many times, these datasets only describe the networks in question in one organism at a time. Furthermore, signaling, metabolic and physical interaction networks are all covered by different types of experiments and data produced by these systems is located in heterogeneous databases. By contrast, phylogenetic profiles can potentially uncover all three types of networks from sequencing data alone. This was highlighted in our work during retrieval of potential interactors within the sexual reproduction and kinetochore networks with the retrieval of LFR and CFAP157, respectively. CFAP157, a cilia and flagella associated protein might be involved in recruitment/regulation of APC-Cdc20 or ciliary kinases (e.g Nek1), both known to mediate APC regulation of ciliary dynamics ([61]). In both cases, a regulatory action within the network was the biological process which involved both the query and retrieved HOGs, not a physical interaction. The advances put forward by our new methodology and the property of

retrieving entire networks and not just physical interactions opens the possibility of performing comparative profiling on an unprecedented scale and lays the groundwork for integrative modeling of the interplay between PPI, regulation and metabolic networks in a more holistic way.

Further work remains to be done on tuning the profile construction with the appropriate weights at each taxonomic level, as well as constructing profiles for subfamiles arising from duplications which may undergo neofunctionalization, a theme which has been previously explored in phylogenetic profiling efforts relying on far fewer genomes [62]. Downstream processing of the explicit representation of the data, as opposed to the hash signature, can also be designed using more computationally intensive methods to detect interactions on smaller subsets of profiles after using the LSH as a first search.

The phylogenetic profiling pipeline presented in this work will be integrated into OMA web-based services. Meanwhile, it is already available on Github as a standalone package (https://github.com/DessimozLab/HogProf).

## Methods

The following section details the creation of phylogenetic profiles using OMA data, their transformation into MinHash based probabilistic data structures and the tools and libraries used in the implementation.

### Profile construction

To generate large-scale gene phylogenies labelled with speciation, duplication and loss events (a.k.a. *enhanced phylogenies* or *tree profiles*) for each HOG in OMA, we processed input data in OrthoXML format [63] with pyHam [24], using the NCBI taxonomic tree [64] pruned to contain only the genomes represented in OMA [16]. Tree profiles contain a species tree annotated at each taxonomic level with information on when the last common ancestor gene appeared, where losses and duplications occurred and the copy number of the gene at each taxonomic level. More information on the pyHam inference of evolutionary events can be found in [24]. pyHam can also be used to infer enhanced phylogenies for other datasets available in OrthoXML format such as ENSEMBL [13] or with data generated from phylogenetic trees such as those found in PANTHER [65] through the use of the function etree2orthoxml() in the tree analysis package ETE3 [66].

The enhanced phylogeny trees for each HOG are parsed to create a vector representation of the presence or absence of a homologue at each extant and ancestral node as well as the duplication or loss events on the branch leading to that node. Each profile vector contains 9345 columns (corresponding to the 3115 nodes of the taxonomy used and the 3 categories of presence, loss and duplication).

To encode profile vectors as weighted MinHash signatures [67] we used the Datasketch library [25]. In this formulation, the Jaccard score between multisets representing profiles can be more heavily influenced by nodes with a higher weight. The final MinHash signatures used were built with 256 hashing functions.

After transforming HOG profile vectors to their corresponding weighted MinHashes using the datasketch library, an estimation of the Jaccard distance between profiles can be obtained by calculating the Hamming distance between their hash signatures [68]. The speed of comparison and lower bound for accuracy of the estimation of the Jaccard score is set by the number of hashing functions. The comparison of hash signatures has $O(N)$ time complexity where N is the number of hash functions used to generate the MinHash signature. Due to this property, an arbitrary number of elements can be encoded in this signature without slowing down

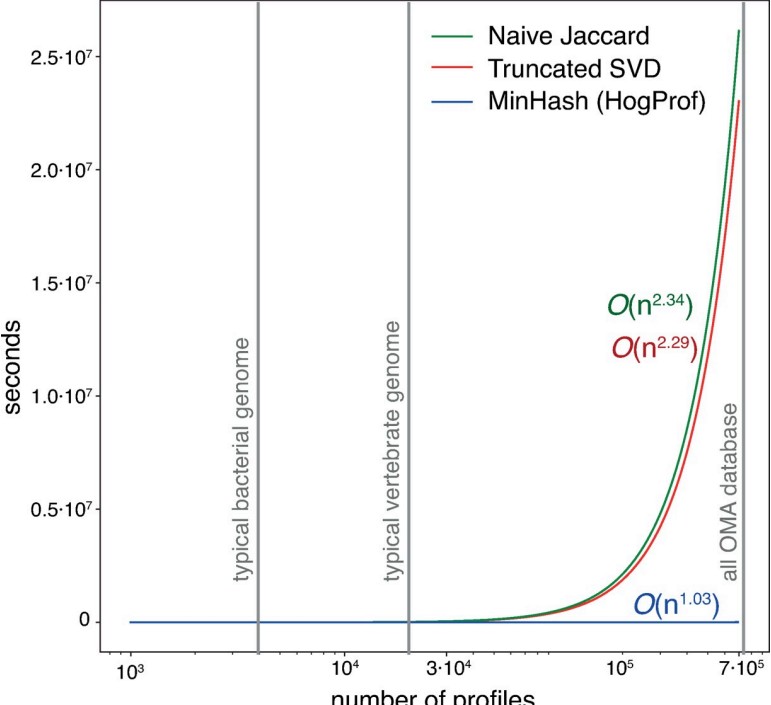

**Fig 6. To illustrate the advantageous scaling properties of MinHash data structures, synthetic profiles of length 100 were generated in the form of binary vectors (0 and 1 equiprobable).** Profiles were then clustered using an explicit calculation of the Jaccard distance, reduced to a lower dimensionality (5 dimensions) with truncated SVD, normalized and explicitly clustered using Euclidean distance as in SVD-Phy [18] or transformed into MinHash signatures and inserted into an LSH Forest object as in our method. Orders of magnitude showing typical use cases for profiling pipelines are shown on the x-axis. Curves were fitted to each set of timepoints to empirically determine the time complexity of each approach.

comparisons. In our use case, this enables the use of an arbitrarily large number of taxa for which we can consider evolutionary events. Additionally, hardware implementations of hash functions allow the calculation of hash signatures at rates of giga hashes per second and allow for extremely fast implementation of this step, placing the bottleneck of the pipeline at the calculation of enhanced phylogenies.

The weighted MinHash objects for each HOG's enhanced phylogeny were compiled into a searchable data structure referred to as a Locality Sensitive Hashing Forest (LSH Forest) [30] and their signatures were stored in an HDF5 file. The LSH Forest can be queried with a hash signature to retrieve the K neighbors with the highest Jaccard similarity to the query hash. The K closest hashes are retrieved from a B-Tree data structure [69]. This branching tree data structure allows for the querying and dynamic insertion and deletion of elements in the LSH Forest data structure built upon it with logarithmic time complexity.

The scaling properties of the MinHash data structures when compared to pairwise distance calculations and hierarchical clustering are shown below in Fig 6.

## Computational resources, data and libraries

Our dataset contains approximately 600,000 HOGs computed from the 2,167 genomes in OMA (June 2018 release) The main computational bottleneck in our pipeline is the calculation of the labelled gene trees for each HOG using pyHam. Even with this computation, compiled LSH forest objects containing the hash signatures of all HOGs' gene trees can be compiled in

under 3 hours (with 10 CPUs but this can scale easily to more cores) with only 2.5 GB of RAM and queried extremely efficiently (an average of 0.01 seconds over 1000 queries against a database containing profiles for all HOGs in OMA on an Intel(R) Xeon(R) CPU E5530 @ 2.40 GHz and 2 GB of RAM to load the LSH database object into memory). This performance makes it possible to provide online search functionality, which we aim to release in an upcoming web-based version of the OMA browser. Meanwhile, the compiled profile database can be used for analysis on typical workstations (note that memory and CPU requirements will depend on the number of hash functions implemented in the construction of profiles and the filtering of the initial dataset to clades of interest to the user).

All gene ontology (GO) annotations (encompassing molecular functions, cellular locations, and biological processes) for HOGs contained in OMA were analyzed with GOATOOLS [70]. To calculate the enrichment of annotations, the results returned by the LSH Forest annotations for all protein sequences contained in the HOGs returned by the search were collected and the entire OMA annotation corpus was used as background.

HDF5 files were compiled with H5PY (ver. 2.9.0). Pandas (ver. 0.24.0) was used for data manipulation. Labelled phylogenies were manipulated with ETE3 [66]. Datasketch (ver. 1.0.0) was used to compile weighted MinHashes and LSH Forest data structures. Plots were generated using matplotlib (ver. 3.0.2). PyHam (ver 1.1.6) was used to calculate labelled phylogenies for the HOGs in OMA.

Time complexity analysis in Fig 6 was done with the scikit-learn implementation of truncated SVD [71] and scipy [72] distance functions.

## Pearson and Spearman correlation comparison of distance matrices

Distance matrices between all pairs of profiles in the kinetochore and APC complex protein families defined in [10] were compared using the Spearman and Pearson statistical analysis functions from the the SciPy python package to verify the monotonicity of the scores between families.

## Supporting information

**S1 Data. Contains the results of GO enrichment analysis done on the results of our search for kinetochore interactors.** After searching with the HOGs corresponding to each of the kinetochore components, the returned HOGs were clustered according to their jaccard similarity. Using a hierarchical clustering and a manually defined cutoff the results were separated into discrete clusters. Each cluster was analyzed using goatools for GO enrichment. Enrichment results for clusters containing a query gene were recorded in this CSV file.
(CSV)

**S2 Data. Contains the goatools output for the GO enrichment analysis of the top 100 closest coevolving HOGs returned by a query with Hap2.**
(CSV)

**S3 Data. Contains the goatools output for the GO enrichment analysis of the top 100 closest coevolving HOGs returned by a query with Gex1.**
(CSV)

**S4 Data. Contains a manually selected set of highly conserved protein families involved in sexual reproduction.**
(CSV)

**S5 Data. Contains the results of GO enrichment analysis done on the results of our search for sexual reproduction network interactors.** After searching with the HOGs corresponding to each of the manually curated list of conserved sexual reproduction network components, the returned HOGs were clustered according to their jaccard similarity. Using a hierarchical clustering and a manually defined cutoff the results were separated into discrete clusters. Each cluster was analyzed using goatools for GO enrichment. Enrichment results for clusters containing a query were recorded in this csv file.
(CSV)

## Acknowledgments

We thank Monique Zahn and the anonymous reviewers for the helpful feedback on the manuscript.

## Author Contributions

**Conceptualization:** David Moi, Pablo S. Aguilar, Christophe Dessimoz.

**Formal analysis:** David Moi, Laurent Kilchoer.

**Funding acquisition:** Pablo S. Aguilar, Christophe Dessimoz.

**Investigation:** David Moi, Christophe Dessimoz.

**Methodology:** David Moi, Pablo S. Aguilar, Christophe Dessimoz.

**Project administration:** Christophe Dessimoz.

**Software:** David Moi, Laurent Kilchoer, Christophe Dessimoz.

**Supervision:** Pablo S. Aguilar, Christophe Dessimoz.

**Validation:** David Moi.

**Visualization:** David Moi.

**Writing – original draft:** David Moi.

**Writing – review & editing:** David Moi, Pablo S. Aguilar, Christophe Dessimoz.

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
