## [Decision Letter · Decision Letter 0]

22 Dec 2019

Dear Dr Dessimoz,

Thank you very much for submitting your manuscript 'Scalable Phylogenetic Profiling using MinHash Uncovers Likely Eukaryotic Sexual Reproduction Genes' for review by PLOS Computational Biology. Your manuscript has been fully evaluated by the PLOS Computational Biology editorial team and in this case also by independent peer reviewers. The reviewers appreciated the attention to an important problem, but raised some substantial concerns about the manuscript as it currently stands. While your manuscript cannot be accepted in its present form, we are willing to consider a revised version in which the issues raised by the reviewers have been adequately addressed. We cannot, of course, promise publication at that time.

Sincerely,

Christos A. Ouzounis

Associate Editor

PLOS Computational Biology

Thomas Lengauer

Methods Editor

PLOS Computational Biology

[LINK]

Reviewer's Responses to Questions

**Comments to the Authors:**

Reviewer #1: In my opinion phylogenetic profiles are one those methods that are intensively researched and developed by computational biologists but relatively poorly utilized by molecular biologist - some notable exceptions of course excluded. The reasons for this relative lack of utilization are many many fold, as also discussed in this manuscript. I sincerely hope that this manuscript will help to close this gap. I do have some comments perhaps not so much on the novel proposed methodology, as more on the way in which the results are introduced and contextualized.

The introduction introduces the initial lack of genome diversity of eukaryotes as one of the issues in adopting phylogenetic profiles for eukaryotes, and then introduces OMA and the HOGs as a nice orthology database with “2000 cellular organisms”. However it is not mentioned how many (and how diverse) eukaryotes OMA contains. It is my impression that the amount and diversity of eukaryotes in OMA is a minority in these 2000 organisms. I think it would be more transparent if the authors explicitly mention the amount (and “diversity”) of eukaryotic organisms in OMA.

The introduction seems to suggest that phylogenetic profiles for many orthology databases are currently not offered. This is not completely true. The STRING-DB still allows phylogenetic profile searches not just on normalized “homology” (by default) but also on orthologs groups (although this option is somewhat hidden).

The introduction argues that the main reason that phylogenetic profiles are not used as much in eukaryotes as they could is speed of similarity computation. Perhaps this is indeed going to be a problem in the near future, but as general assertion I am not entirely convinced this statement is fully true. In our work we have sofar been easily able on our local (admittedly beefy) workstations to successfully compute phylogenetic profile similarity for large eukaryotic data sets. Perhaps this point could be more made strongly if the present manuscript would include a smart implementation of jaccard of profile similarities on simple OMA/HOG presence/absence profile and show that indeed how/where the computational bottleneck is. (or perhaps the manuscript already present such an analysis and I missed it).

I think that the orthology database and the method of phylogenetic profile searching are not strictly necessarily connected. The introduced MinHash search method seems to need an orthology that allows a species tree to be annotated with duplications and losses. Such data are available elsewhere. Most easily they should be extractable from the PANTHER database. But also EGGNOG is hierarchical and they could perhaps also be retrieved from numerous ENSEMBL compara genome subsets. I think it would strengthen the message of applicability of this method if it would be applied to other orthology datasets.

For evaluating potential novel connections to kinetochore it appears the proteins detailed in Table 2 exemplify another problem with finding wide-spread utilization of phylogenetic profiles by molecular biologists. So I reached out via the bioRxiv version of this article to a molecular biologist somewhat familiar with the kinetochore. It seems that the co-evolution of APC12 with CDC26 is a spurious orthology/identifier problem as CDC26 is a synonym of APC12 and reference [29] used as evidence still using the old nomenclature for APC12. The co-evolution of KNL1 with TACC3 is asserted to bind to the kinetochore but insofar as they understand the literature this is not the case and reference [30] is also not showing that. Some very indirect linkage of TACC3 to kinetochore function is known to the extent that TACC3 is microtubule-associated and seems to be stabilizing the spindle, but that does not qualify as being part of the well defined set of complexes that make up the kinetochore. The other links were seen as not specific enough to be relevant for a molecular biologists but I guess this dismissal by experimentalist is more a Gene Ontology versus real biology problem than something inherent to phylogenetic profiles.

In the discussion, potential expansions of this method to account for neofunctionalization after duplications are mentioned. This is indeed one of those cool and difficult things on thinking about phylogenetic profiles and the evolution of function. When discussing this possible extension of the method it could be worth to add another citation to an already extensive citation list. Because this paper: doi: 10.1016/j.celrep.2015.01.025 from Tobias Meyer already makes phylogenetic profile searches where the neofunctionalization is explicitly taken into account.

Reviewer #2: Dear Authors.

Please see some of my comments on the annotated pdf attached. Although you have presented a study with potential relevance in bioinformatics and computational biology, I consider that the ms needs heavy revisions to accomplish the criteria for publication in the Journal.

Thanks,

**Have all data underlying the figures and results presented in the manuscript been provided?**

Reviewer #1: Yes

Reviewer #2: Yes

PLOS authors have the option to publish the peer review history of their article (what does this mean?). If published, this will include your full peer review and any attached files.

Reviewer #1: No

Reviewer #2: No

---

## [Decision Letter · Decision Letter 1]

15 Apr 2020

Dear Dr. Dessimoz,

Thank you very much for submitting your manuscript "Scalable Phylogenetic Profiling using MinHash Uncovers Likely Eukaryotic Sexual Reproduction Genes" for consideration at PLOS Computational Biology. As with all papers reviewed by the journal, your manuscript was reviewed by members of the editorial board and by several independent reviewers. The reviewers appreciated the attention to an important topic. Based on the reviews, we are likely to accept this manuscript for publication, providing that you modify the manuscript according to the review recommendations.

Sincerely,

Christos A. Ouzounis

Associate Editor

PLOS Computational Biology

Thomas Lengauer

Methods Editor

PLOS Computational Biology

[LINK]

Reviewer's Responses to Questions

**Comments to the Authors:**

Reviewer #1: The authors have extensively discussed the suggestions and edited the manuscript to accommodate them.

Reviewer #2: Please see some minor comments on the attached pdf file. I am very happy with the changes on the ms.

**Have all data underlying the figures and results presented in the manuscript been provided?**

Reviewer #1: Yes

Reviewer #2: Yes

PLOS authors have the option to publish the peer review history of their article (what does this mean?). If published, this will include your full peer review and any attached files.

Reviewer #1: No

Reviewer #2: No
---

## [Decision Letter · Decision Letter 2]

18 May 2020

Dear Dr. Dessimoz,

We are pleased to inform you that your manuscript 'Scalable Phylogenetic Profiling using MinHash Uncovers Likely Eukaryotic Sexual Reproduction Genes' has been provisionally accepted for publication in PLOS Computational Biology.

Best regards,

Christos A. Ouzounis

Associate Editor

PLOS Computational Biology

Thomas Lengauer

Methods Editor

PLOS Computational Biology

Reviewer's Responses to Questions

Comments to the Authors:

Please note here if the review is uploaded as an attachment.

Reviewer #1: No further comments.

Have all data underlying the figures and results presented in the manuscript been provided?

Large-scale datasets should be made available via a public repository as described in the 

PLOS Computational Biology

data availability policy, and numerical data that underlies graphs or summary statistics should be provided in spreadsheet form as supporting information.

Reviewer #1: Yes

PLOS authors have the option to publish the peer review history of their article (what does this mean?). If published, this will include your full peer review and any attached files.

Do you want your identity to be public for this peer review?

 For information about this choice, including consent withdrawal, please see our Privacy Policy.

Reviewer #1: No

---

## [Editor Report · Acceptance letter]

6 Jul 2020

PCOMPBIOL-D-19-01799R2 

Scalable Phylogenetic Profiling using MinHash Uncovers Likely Eukaryotic Sexual Reproduction Genes

Dear Dr Dessimoz,

I am pleased to inform you that your manuscript has been formally accepted for publication in PLOS Computational Biology. Your manuscript is now with our production department and you will be notified of the publication date in due course.

With kind regards,

Laura Mallard
